# Polyelectrolyte Complexes of Natural Polymers and Their Biomedical Applications

**DOI:** 10.3390/polym11040672

**Published:** 2019-04-12

**Authors:** Masayuki Ishihara, Satoko Kishimoto, Shingo Nakamura, Yoko Sato, Hidemi Hattori

**Affiliations:** 1Division of Biomedical Engineering Research Institute, National Defense Medical College, Saitama 359-8513, Japan; snaka@ndmc.ac.jp (S.N.); ysato@ndmc.ac.jp (Y.S.); 2Research Support Center, Dokkyo Medical University, Tochigi 321-0293, Japan; skishi@dokkyomed.ac.jp; 3Department of Biochemistry and Applied Sciences, University of Miyazaki, Miyazaki 889-2162, Japan; h-hattori@cc.miyazaki-u.ac.jp

**Keywords:** glycosaminoglycan, chitin/chitosan, polyelectrolyte complexes, cell delivery carrier, drug delivery carriers

## Abstract

Polyelectrolyte complexes (PECs), composed of natural and biodegradable polymers, (such as positively charged chitosan or protamine and negatively charged glycosaminoglycans (GAGs)) have attracted attention as hydrogels, films, hydrocolloids, and nano-/micro-particles (N/MPs) for biomedical applications. This is due to their biocompatibility and biological activities. These PECs have been used as drug and cell delivery carriers, hemostats, wound dressings, tissue adhesives, and scaffolds for tissue engineering. In addition to their comprehensive review, this review describes our original studies and provides an overview of the characteristics of chitosan-based hydrogel, including photo-cross-linkable chitosan hydrogel and hydrocolloidal PECs, as well as molecular-weight heparin (LH)/positively charged protamine (P) N/MPs. These are generated by electrostatic interactions between negatively charged LH and positively charged P together with their potential biomedical applications.

## 1. Introduction

Chitin is the second-most abundant natural polysaccharide after cellulose and is composed of *N*-acetylglucosamine. Chitin is primarily produced from crustaceous exoskeletons, a waste product of the seafood industry that would otherwise pollute coastal areas. Chitin is generated through the dissolution of calcium carbonate and the removal of proteins. Chitosan is produced by full or partial deacetylation of the *N*-acetylglucosamine residues in chitin. Chitosan is composed of co-polymers of glucosamine and *N*-acetylglucosamine linked by β(1→4) glycosidic bonds, generated by partial alkaline deacetylation of chitin [1,2]. Chitin and chitosan are biocompatible with living tissue and are nontoxic [3,4]. The term “chitosan” generally refers to a natural cationic biopolymer comprising less than 50% *N*-acetylglucosamine residues [2,3,4]. The solubility, hydrophobicity, and electrostatic properties of chitosan depend on the degree of deacetylation and on the molecular weight [5]. Chitosan is hydrolyzed by lysozyme and therefore is a biodegradable polymer. The degradation products of chitosan are biocompatible, noncarcinogenic, and nonimunogenic [4,5,6,7], and exhibit biological activity such as antimicrobial activities [8], hypocholesterolemic functions [9], antitumor activities [10,11], and stimulatory functions in wound healing [12,13].

In contrast, glycosaminoglycans (GAGs) are polysaccharide chains, such as heparinoids (heparin/heparin sulfate (HS) and other heparin-like molecules), chondroitin sulfate, dermatan sulfate, and keratan sulfate. All bear negative charges that vary in density and position [14,15]. Heparinoids interact with a variety of functional proteins, including heparin-binding growth factors (GFs), cytokines, extracellular matrix components, and adhesion molecules [16,17]. Most biological functions of heparinoids depend on the binding of these functional proteins to the polysaccharide chains, mediated by specific domains, with distinct saccharide sequences [18,19,20]. For example, interactions of heparinoids with fibroblast growth factor (FGF)-1 and FGF-2 require different combinations of sulfate groups, and thus require different saccharide sequences [21,22,23,24]. Furthermore, specific heparinoid structures interact with hepatocyte growth factor [25,26] and vascular endothelial growth factor (VEGF) [27].

Electrostatic interactions between oppositely charged polyelectrolytes generate polyelectrolyte complexes (PECs) [28,29]. Both synthetic and natural PECs can interact with proteins [30,31]. PECs with protein-binding characteristics can be used to study the behavior of polyelectrolytes. Interactions of PECs with proteins in liquid result in dispersions, soluble complexes, emulsification, and/or the formation of amorphous precipitates. Many studies have reported the chemical properties of PECs, obtained under various experimental conditions, such as the strength and position of ionic sites, charge density, and rigidity of the polymer chains [32,33,34].

Mechanisms, critical experimental aspects, and applications on PECs were comprehensively reviewed [35]. This review focuses on PEC hydrogels formed by the chemical interaction of chitosan and crosslinkers [36], such as photo-crosslinked chitosan hydrogel (PCH) formed by the addition of a photocrosslinker [37,38], ionically crosslinked chitosan hydrogels [34,36], temperature sensitive chitosan hydrogels [39,40], and hydrocolloids [41,42] formed by direct interaction between polymeric chains without addition of a crosslinker. Furthermore, we will discuss the potential medical applications of PCH [37,38] and chitosan-based biomaterials, such as alginate/chitosan/fucoidan complexed hydrocolloid sheets (ACF-HSs) [41,42] and PECs [43,44,45]. We previously studied ACF-HSs and PECs as drug delivery carriers, cell delivery carriers, tissue adhesives, wound dressings, hematostats, scaffolds for tissue engineering, and protein/gene delivery carriers.

## 2. Chitosan-Based PEG Hydrogels

Chitosan is only soluble in acidic solvents such as diluted hydrochloric acid, acetic acid, propionic acid, and ascorbic acid, making this reagent difficult to handle as a biomaterial in wound dressings and biological adhesion treatments [46]. Numerous studies have attempted to improve the water solubility of chitosan over a broad pH range to provide the polymer with advanced functionalities; modifications have included changes to carbohydrate branching and derivatization using different types of disaccharides, including lactose, maltose, and cellobiose [46]. Based on the definition for PECs as versatile formulations formed by electrostatic interactions between oppositely charged biopolymers, chitosan (and the derivatives)-based hydrogel can be a member of PECs. Chitosan-based PEC hydrogels have been defined as networks of crosslinked and hydrated chitosan. Chitosan hydrogels are classified into two groups according to this definition, namely, chemical chitosan hydrogels formed by irreversible covalent bonds, and physical chitosan hydrogels formed by various reversible bonds [31,32]. Physical chitosan hydrogels include ionically crosslinked colloids, PECs [43,44,45], and ACF-HS comprising alginate, chitosan and fucoidan [41,42]. In contrast, in chemical chitosan hydrogels, chitosan and its derivatives are covalently interconnected by low molecular weight crosslinkers, leading to the formation of a three-dimensional (3D) hydrated network. Crosslinking density, which is determined by the molar ratio of crosslinkers to the repeating units in chitosan and its derivatives, affects the properties of chemically crosslinked chitosan hydrogels [47,48].

Preparation of a covalently crosslinked chitosan hydrogel requires the use of crosslinkers with two or more reactive groups to crosslink chitosan chains, such as glutaraldehyde as a dialdehyde [36] (Figure 1A). However, it is difficult to completely eliminate free unreacted dialdehydes in hydrogels, which may induce toxic effects. Figure 1B shows a simplified scheme for photocrosslinked chitosan hydrogels that form upon exposure to visible or ultraviolet light in the presence of photocrosslinkers [37,38] (Figure 1B).

Several reports have described temperature-sensitive chitosan hydrogels that show a sol-gel transition due to a conformational change at 37 °C. Since chitosan lacks intrinsic thermosensitive properties, the introduction of temperature-sensitive materials is required to make a temperature-sensitive chitosan hydrogel. For example, temperature-sensitive hydrogels composed of chitosan and β-glycerophosphate [39,40] or polyethylene glycol (PEG) [36] have been prepared and investigated for their sol-gel transition in response to thermal and pH changes. These hydrogels were evaluated as carriers for cells and functional proteins [46,47].

A photocrosslinkable chitosan derivative (Az-CH-LA), that contains both lactobionic acid and photoreactive *p*-azidobenzoic acid, used as a photocrosslinker [37,38]. Chitosan used in this study had a molecular weight of 300–500 kDa, with 80% degree of deacetylation. Lactose moieties have been introduced through a condensation reaction with amino groups of the chitosan. The chitosan to which 2% lactobionic acid was introduced (CH-LA) exhibited a good aqueous solubility up to 3 w% at even neutral pH. Furthermore, the application of ultraviolet light (UV) irradiation, at a lamp distance of 2 cm (Spot Cure ML-251C/A with a guide fiber unit (SF-101BQ) and 250 W lamp (240–380 nm; major peak: 340 nm), Usio Electrics Co., Ltd., Tokyo, Japan) to Az-CH-LA to which 2.5% p-azidebenzoic acid, was introduced to produce an insoluble hydrogel within 30 s and firmly held together two pieces of porcine tissue [37,38]. A 3% or lower Az-CH-LA solution can be injected into tissue and an insoluble hydrogel is formed following external irradiation with a UV light. Furthermore, other types of chitosan hydrogels have been reported as biomaterials, including crosslinked polydopamine/nanocellulose hydrogels [49], temperature-sensitive chitosan-based injectable hydrogels [50], PEG poly (L-alanine) thermogels [51], hyaluronate/alginate hydrogels [52], pH-responsive tannic acid-carboxylated agarose composite hydrogels [53], catechol-functionalized chitosan/pluronic hydrogels [54], and UV crosslinked biodegradable gelatin [55]. All these hydrogels have potential applications for wound healing, as tissue adhesives, and in tissue engineering.

## 3. Applications of Chitosan-Based PEC Hydrogels for Wound Healing

Chitosan promotes rapid dermal regeneration and accelerates wound healing by stimulating macrophages and acting as a chemoattractant for neutrophils, an early event essential in wound healing [13,56]. Neutrophils kill microorganisms, remove dead cells, and stimulate other immune system cells, thereby improving overall healing by reducing the opportunity for infection [57]. Chitosan and its derivatives have been applied as a dressing for wound healing. Water-soluble chitin/chitosan (WSC) solution exhibits higher tensile strength and promotes faster healing than insoluble chitin and chitosan powders. It is possible that the higher biodegradability and hydrophilicity of WSC solutions, compared to powders, increase its biocompatibility and wound healing stimulatory activity [57]. Furthermore, chitosan has been combined with several functional molecules such as cytokines, GFs, and extracellular matrix components to improve the healing process. Chitosan enhances the healing of decubitus ulcers and wounded meniscal tissues, and depresses scar formation and retraction during healing [57].

The application of PCH to open wounds in normal mouse [58,59] and rat [12,60] induces remarkable wound contraction, thereby accelerating the wound closure and healing processes. In addition, PCH can exhibit sustained release of angiogenic GFs, thereby acting as a carrier and promoting vascularization, in vivo. For example, fibroblast growth factor 2 (FGF-2) interacts with chitosan molecules in PCH and is gradually released from PCH during in vivo biodegradation [61]. FGF-2-incorporated PCH (FGF-2&PCH) induces substantial vascularization and granulation tissue formation, and improves wound healing in healing-impaired diabetic *db*/*db* mice (Figure 2) [62]. Interestingly, FGF-2&PCH has only a minor effect on healing in normal *db*/+ littermate mice. Although the cause of this minor effect is not completely understood, it is likely that the presence of sufficient macrophages has a significant effect on the formation of wound granulation tissue in *db*/+ mice [63]. Furthermore, poor wound healing in *db*/*db* mice may be explained by a defect in VEGF expression [64]. In other words, *db*/+ mice may have sufficient VEGF for angiogenesis and wound repair.

## 4. Glycosaminoglycan (GAG)-Based PECs

Acidic polymers, such as GAGs, complexed with basic polymers, such as protamine and chitosan, form PECs through electrostatic interactions [44,45,46]. Several reports demonstrate that the acidic and basic polymers in PECs bind to various proteins such as cytokines and GFs above and below the isoelectric points of the proteins. These interactions generate various forms, such as nanoparticles, colloidal particles, soluble complexes, and amorphous precipitates (Figure 3). The electrostatic interactions between charged polymers are very interesting due to their similarity to biological interactions. For example, interactions between nucleic acids and proteins play an important role in transcription processes [65]. DNA/chitosan PECs [65], chitosan/hyaluronate PECs, and chitosan/chondroitin sulfate PECs were reported to function as gene and/or protein carriers [65,66]. Furthermore, PECs that are insoluble also have potential applications, such as hydrogels, sheets, microcapsules, and scaffolds for tissue regeneration [67].

CH-LAs water soluble at neutral pH have been prepared by the introduction of lactobionic acid into aqueous solutions of acidic polymers [37,38,46]. A 2 wt% aqueous solution of CH-LA is highly viscous and readily gels when mixed with solutions of acidic polymers such as GAGs, and especially when mixed with heparinoids (heparin and heparin-like molecules). The products are injectable hydrogels, due to the polyelectrolytic interaction between basic CH-LA and acidic heparinoids, such as non-anticoagulant (NAC)-heparin [68,69], 6-*O*-desulfated heparin [70], and fucoidan [71]. When NAC-heparin/CH-LA hydrogel, containing FGF-2 was subcutaneously injected into the backs of rats or mice, fibrous tissue formation and neovascularization were markedly enhanced around the injection sites. Furthermore, gradual release of FGF-2 from the hydrogel stimulated collateral circulation and angiogenesis [68,69]. Conversely, PCH containing paclitaxel, an inhibitor of angiogenesis and an anti-cancer agent, effectively suppressed angiogenesis and tumor growth in mice [72].

## 5. Applications of GAG-Based PECs

The biological properties and applications of GAG-based materials have been investigated [73]. GAGs, containing heparinoids, have been deposited on the surfaces of micro/nanoparticles made of magnets [74], metals [75,76], synthetic polymers [77,78,79], and natural biopolymers [80,81]. The combination of functional molecules such as proteins (e.g., GFs) and DNA with these micro-/nano-particles result in particles with potential applications in various biomedical fields [73]. Micro-/nano-particles can be passivated by combining with GAG, improving their biocompatibility. Polysaccharides, found on the surfaces of all eukaryotic cells, are another interesting family of molecules for coating foreign materials.

Electrostatic interactions between oppositely charged polyelectrolytes generate polyelectrolyte complexes, such as low-molecular-weight heparin (LH) (Fragmin) and protamine (P) [43,44]. Non-stoichiometric PECs carry excess charge when this interaction occurs non-equivalently [80,81,82]. PECs interact with and adsorb various functional proteins [83]. Heparinoids such as LH (MW: approximately 5000 Da) specifically bind to various proteins such as cytokines, GFs, extracellular matrix, and adhesion molecules with high affinity [21,67]. In this way, heparinoids may be beneficial as pharmaceuticals for treating various pathological conditions. However, the use of high-dose heparin carries an excessive risk of bleeding [84]. In contrast, LH offers practical and pharmacological advantages compared to heparin. The lower binding affinity of LH to heparin-binding coagulation factors lead to a low and predictable anticoagulant response and thus laboratory monitoring of drug levels is not necessary to adjust the dosage [84]. Furthermore, one or two subcutaneous injections per day are sufficient to maintain therapeutic concentrations due to the long plasma half-life of LH [84]. On the other hand, protamine, a mixture of basic proteins produced from fish sperm, neutralizes the blood anticoagulant activity of heparin and LH by forming a stable complex that lacks anticoagulant activity [85]. Protamine (P) is used clinically as an antagonist for the anticoagulant activity of heparin to treat heparin-induced bleeding [86]. As shown in Figure 4, PECs comprising low molecular weight heparin/protamine nano/micro particles (LH/P N/MPs) were generated by mixing LH with P at a ratio of 6:4 [43,44]. LH/P N/MPs are 0.1–3 μm in diameter and specifically bind FGF-2 [43,44], HGF [45], and other GFs secreted from platelets and can protect and activate various GFs. [87]. In addition, LH/P N/MPs are adsorbed onto cell surfaces and ECM in various tissues. GF-containing LH/P N/MPs could significantly induce fibrous tissue formation and vascularization by stabilizing, activating, and gradually releasing GFs from GF-containing LH/P N/MPs [67,87].

The multipotent characteristics of adipose-derived stromal cells (ADSCs), as well as their abundance in the human body, make them an attractive potential resource for wound repair and tissue engineering. ADSC transplantation has been used in combination with biomaterials, including cell sheets, hydrogels, and 3D scaffolds [88], making the use of ADSCs an option for regenerative medicine [89]. ADSCs efficiently proliferate in a 3D culture medium comprising 1–6% human serum and Dulbecco’s modified Eagle’s medium (DMEM) mixed with a gel supplemented with LH/P N/MPs [67]. Many LH/P N/MPs bind to the surfaces of adhesive cells, including ADSCs and tumor cells, through interactions with cell surface heparin-binding proteins such as integrins [67,90]. The interaction of LH/P N/MPs with cell surfaces leads to formation of a cell-LH/P N/MP-aggregate within two hours (Figure 5A). Cell viability significantly increases in such in vitro aggregates [90,91]. Injection of cell-LH/P N/MP-aggregates induced fibrous tissue formation and vascularization in vivo [90,91]. Thus, LH/P N/MPs can function as cell carriers that increase cell viability.

As shown in Figure 5B, ADSCs efficiently proliferate in 3D culture using a 2% human plasma-DMEM gel, containing LMWH/P N/MPs [91]. Inbred rat (IR-) ADSCs were cultured in plasma gel consisting of inbred rat plasma (IRP) (6%)-DMEM gel with FGF-2-containing LH/P N/MPs, then applied to full thickness skin wounds on the backs of streptozotocin-induced diabetic rats [91] (Figure 5C). Wounds treated with ADSCs in IRP-DMEM gel, with FGF-2-containing LH/P N/MPs, closed and healed significantly faster than untreated wounds [91]. The histological examination of wounds treated with IR-ADSCs showed significantly enhanced granulation tissue formation, epithelialization, and capillary formation. These results suggest that some of the transplanted IR-ADSCs had been taken up by the skin tissues and promoted wound healing [91], and that IR-ADSCs grown in IRP-DMEM gel are effective for repairing healing-impaired wounds in diabetic rats.

## 6. Conclusions

Polyelectrolyte complex hydrogels, composed of chitosan and GAG, have attracted considerable attention, due to their compatibility and biological activities, such as wound healing, antimicrobial and antitumor activities, and hypocholesterolemic functions. This review article described PEC hydrogels, generated using polysaccharides, such as positively charged chitosan and negatively charged GAGs such as heparinoids and their derivatives. Photocrosslinked chitosan hydrogels (PCHs) form on exposure to visible or ultraviolet light in the presence of photocrosslinkers and interact with negatively charged heparinoids and various GFs, cytokines, and adhesive molecules. The complexes show promise as functional hemostats and as wound dressings.

The beneficial effects of GAGs, especially heparinoids, can be exploited by incorporation into drug delivery systems. In addition to their anticoagulant action, heparinoids are associated with various cytokine and GF biological processes. Heparinoids are highly soluble and dispersible in water and thus their use often requires an adequate medium, such as hydrogels or PECs to adsorb or retain the complexes. PECs, comprising protamine mixed with LH/P N/MPs, composed of heparinoids and chitosan, have been studied for biomedical applications, such as drug and cell delivery carriers. Thus, biomaterials comprising polysaccharide-based composite PEC hydrogels and N/MPs have potential for many medical applications.

## Figures and Tables

**Figure 1 polymers-11-00672-f001:**
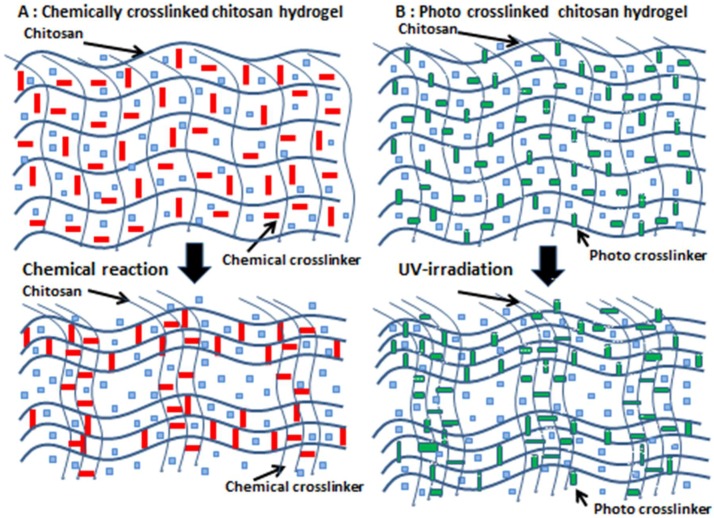
Simplified schemes of gelling mechanisms. (**A**): Chemical crosslinking gelation due to chemical reactions between crosslinkers and polymers. (**B**): Photocrosslinking due to radical reactions between photocrosslinkers and polymers.

**Figure 2 polymers-11-00672-f002:**
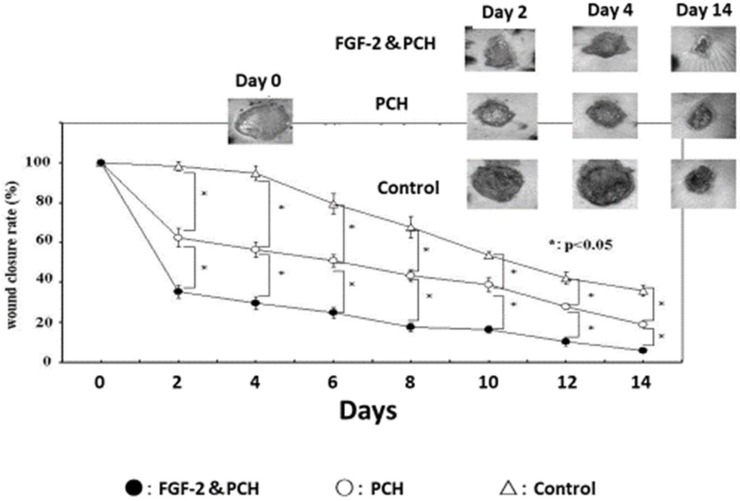
Enhanced wound healing in FGF-2&PCH-treated *db*/*db* mice. FGF-2&PCH stimulates wound healing in diabetic *db*/*db* mice by the synergistic effects of PCH and FGF-2.

**Figure 3 polymers-11-00672-f003:**
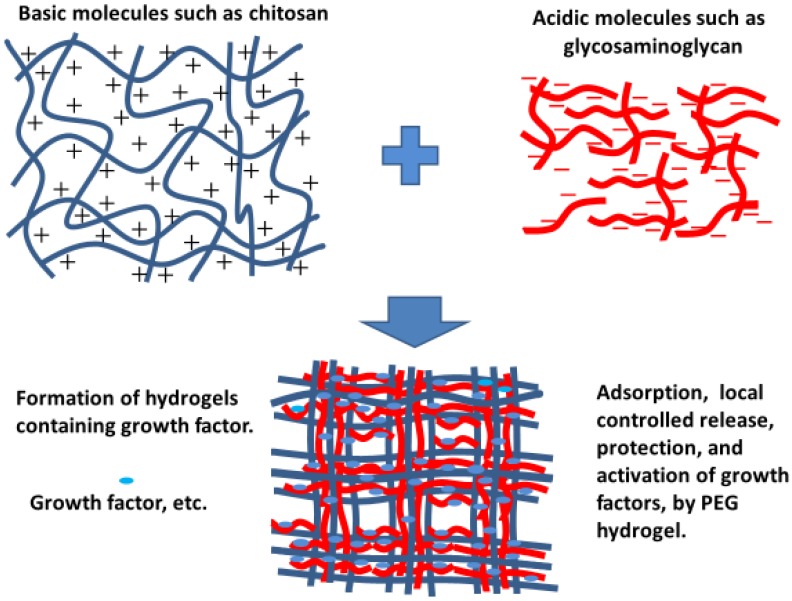
Polyelectrolyte complexes (PECs) comprising acidic and basic polymers.

**Figure 4 polymers-11-00672-f004:**
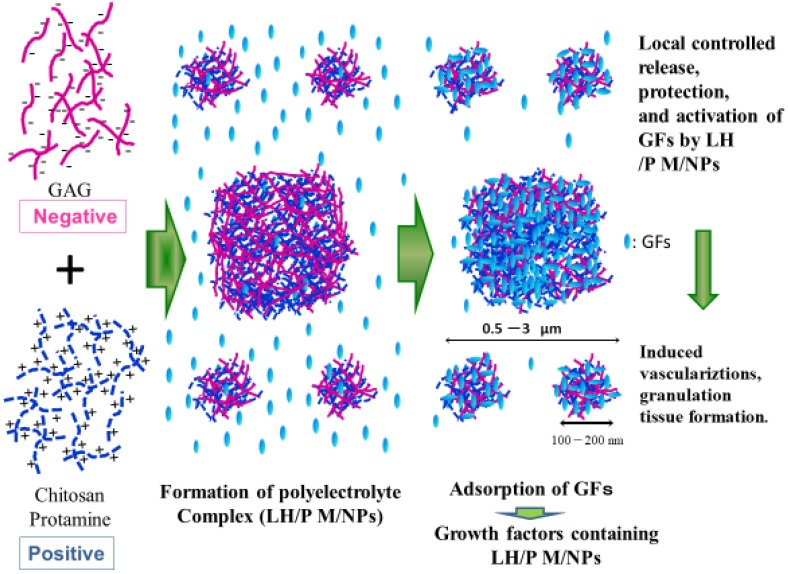
Production of growth factor (GF)-containing LH/P N/MPs as PECs. LH/P N/MPs are specifically bound to FGF-2, HGF, and other GFs secreted from platelets.

**Figure 5 polymers-11-00672-f005:**
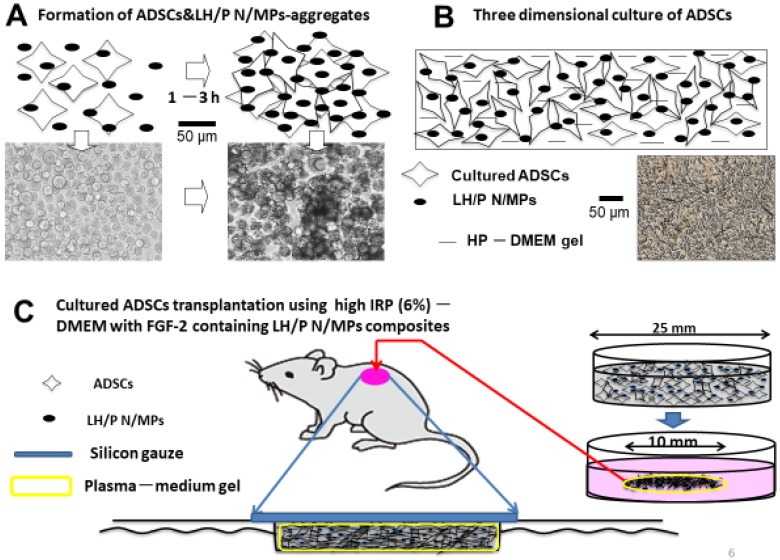
(**A**): Generation of adipose-derived stromal cells (ADSC)-aggregates. (**B**): Three-Dimensional (3D) culture. (**C**): Transplantation of 3D-cultured ADSCs.

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
