# Peer review of "Polyelectrolyte Complexes of Natural Polymers and Their Biomedical Applications"

_polymers, 2019, doi:10.3390/polym11040672_

Round 1
Reviewer 1 Report
The work by Ishihara et al. reports a literature survey on the design and applications of polyelectrolyte complexes. The work is very well written and does not require substantial style and language editing. However, for what concerns the content of the review, the work is not suitable for publication in Polymers in its current version and requires some major revisions. A list of suggestions will follows:
1- in the abstract the authors states that “this review focuses on our original studies”. Focusing a literature review on authors’ own works is somehow a limitation, as a good survey should summarize the main work on a specific topic reported in literature.
2- the authors should highlight the main strength and novelty of their review compared to those already published, such as
- “Polyelectrolyte Complexes (PECs) for Biomedical Applications” by Buriuli M. and Verma D., 2017
- “Polyelectrolyte complexes of chitosan: formation, properties and applications” by M A Krayukhina, N A Samoilova and I A Yamskov, 2008
- “Chitosan-based polyelectrolyte complexes: A review” by A. V. Il’ina V. P. Varlamov, 2005
- “Polyelectrolyte complexes: mechanisms, critical experimental aspects, and applications” by Abhijeet D. Kulkarni et al., 2016
3- Polyelectrolyte complexes (PECs) are defined as versatile formulations formed by electrostatic interactions between oppositely charged biopolymers. Based on this definition that also the authors report, the rationale underpinning authors’ choice to draft a paragraph on physical and chemical hydrogels based on pure chitosan should be explained. Additionally, in this paragraph (paragraph 2) the authors only approximately explain the mechanisms which induce the formation of chemically cross-linked chitosan hydrogels (either via photoirradiation or through cross-linking agents) and temperature-sensitive chitosan hydrogels.
4- On page 3, lines 100-103, further details on photo-polymerization protocol (e.g., power density, UV wavelength) and polymer solubility in neutral pH (details on the meaning of “high”) should be added.
5- Paragraph 4 should be divided in two sub-sections, one focused on the definition and composition of PECs based on acidic and basic polymers and another focused on their application in the biomedical field.
Author Response
The work by Ishihara et al. reports a literature survey on the design and applications of polyelectrolyte complexes. The work is very well written and does not require substantial style and language editing. However, for what concerns the content of the review, the work is not suitable for publication in Polymers in its current version and requires some major revisions. A list of suggestions will follows:
1- in the abstract the authors states that “this review focuses on our original studies”. Focusing a literature review on authors’ own works is somehow a limitation, as a good survey should summarize the main work on a specific topic reported in literature.
There are several published review articles on PECs and their applications. In fact, although we submitted a comprehensive review on PECs and their applications last year, it was rejected because o lack of originality. In this review, we tried to comprehensively describe on PECs of natural polymers、especially natural polysaccharides and their biomedical applications and on our original studies. Abstract was revised to show aspects of this review article
“In addition to their comprehensive review, this review describes on our original studies---”
2- the authors should highlight the main strength and novelty of their review compared to those already published, such as
- “Polyelectrolyte Complexes (PECs) for Biomedical Applications” by Buriuli M. and Verma D., 2017
The paper described comprehensive review on PECs focusing mechanism of PEC formation, structural aspect, physicochemical characterization, and application. In the paper, PECs was classified into three type: soluble, colloidally stable, and coacervated complexes. Furthermore, the paper avoided the use of various chemical cross-linking agents in preparation of PECs. It did not describe on hydrogel such as photocrosslinkable chitosan hydrogel which are main aspects in this review.
- “Polyelectrolyte complexes of chitosan: formation, properties and applications” by M A Krayukhina, N A Samoilova and I A Yamskov, 2008
The paper described comprehensive summary of studies of polyelectrolyte complexes of chitosan and their application. It did not describe on modified chitosan derivative such as photocrosslinkable chitosan and low molecular weight heparin/protamine nano/micro particles (LH/P N/MPs) which are main aspects in this review.
- “Chitosan-based polyelectrolyte complexes: A review” by A. V. Il’ina V. P. Varlamov, 2005
The paper described comprehensive summary of studies of polyelectrolyte complexes of chitosan and their application. It did not describe on modified chitosan derivative such as photocrosslinkable chitosan and low molecular weight heparin/protamine nano/micro particles (LH/P N/MPs) which are main aspects in this review.
- “Polyelectrolyte complexes: mechanisms, critical experimental aspects, and applications” by Abhijeet D. Kulkarni et al., 2016
The paper described comprehensive review on PECs focusing mechanism of PEC formation, structural aspect, preparation, physicochemical characterization, and application. It did not describe on modified chitosan derivative such as photocrosslinkable chitosan and low molecular weight heparin/protamine nano/micro particles (LH/P N/MPs) which are main aspects in this review. To describe it, we added one sentence with ref.35 (page 2, lines 59 – 60).
3- Polyelectrolyte complexes (PECs) are defined as versatile formulations formed by electrostatic interactions between oppositely charged biopolymers. Based on this definition that also the authors report, the rationale underpinning authors’ choice to draft a paragraph on physical and chemical hydrogels based on pure chitosan should be explained. Additionally, in this paragraph (paragraph 2) the authors only approximately explain the mechanisms which induce the formation of chemically cross-linked chitosan hydrogels (either via photoirradiation or through cross-linking agents) and temperature-sensitive chitosan hydrogels.
We appreciated this suggestion. We added following paragraph, “Chitosan is only soluble in acidic solvents such as diluted hydrochloric acid, acetic acid, propionic acid, and ascorbic acid, making this reagent difficult to handle as a biomaterial in wound dressings and biological adhesion treatments. Numerous studies have attempted to improve the water solubility of chitosan over a broad pH range to provide the polymer with advanced functionalities; modifications have included changes to carbohydrate branching and derivatization using different types of disaccharides, including lactose, maltose, and cellobiose. Based on the definition for PECs as versatile formulations formed by electrostatic interactions between oppositely charged biopolymers, chitosan (and the derivatives)-based hydrogel can be a member of PECs.” with ref. 64 (page 2, lines 70 - 77).
4- On page 3, lines 100-103, further details on photo-polymerization protocol (e.g., power density, UV wavelength) and polymer solubility in neutral pH (details on the meaning of “high”) should be added.
We added details on photo-polymerization protocol (e.g., power density, UV wavelength) and polymer solubility in neutral pH (page 3, lines 104 - 112). Although the “high” meant high solubility, the expression was modified to “a good aqueous solubility up to 3 w% at even neutral pH” (page 3, lines 108 - 109).
5- Paragraph 4 should be divided in two sub-sections, one focused on the definition and composition of PECs based on acidic and basic polymers and another focused on their application in the biomedical field.
We agreed with the reviewer`s suggestion. We divided the paragraph 4 into two sub-section according to the reviewer’s suggestion.
New paragraph 4: 4. Glycosaminoglycan (GAG)-based PECs.
New paragraph 5: 5. Applications of GAG-based PECs

Reviewer 2 Report
An interesting article, the abstract needs to be reviewed taking into account the aspects presented and reviewed in the article.
Author Response
An interesting article, the abstract needs to be reviewed taking into account the aspects presented and reviewed in the article.
We revised the abstract (red, underlined) to make the aspects clear according to the reviewer’s comment.

Round 2
Reviewer 1 Report
The authors have significantly improved the quality of the work, extensively answering to all reviewer's comments. The work is thus suitable for publication in Polymers in its current form